# An Optimized Checkerboard Method for Phage-Antibiotic Synergy Detection

**DOI:** 10.3390/v14071542

**Published:** 2022-07-14

**Authors:** Isidora Nikolic, Darija Vukovic, Damir Gavric, Jelena Cvetanovic, Verica Aleksic Sabo, Sonja Gostimirovic, Jelena Narancic, Petar Knezevic

**Affiliations:** PK Laboratory, Department of Biology and Ecology, Faculty of Sciences, Trg Dositeja Obradovica 3, University of Novi Sad, 21000 Novi Sad, Serbia; isidoran@dbe.uns.ac.rs (I.N.); darija.vukovic@dbe.uns.ac.rs (D.V.); damir.gavric@dbe.uns.ac.rs (D.G.); jelena.cvetanovic@dbe.uns.ac.rs (J.C.); verica.aleksic@dbe.uns.ac.rs (V.A.S.); sonja.gostimirovic@dbe.uns.ac.rs (S.G.); jelenan@dbe.uns.ac.rs (J.N.)

**Keywords:** phage-antibiotic synergy, checkerboard method, time-kill curve

## Abstract

Phage-antibiotic synergy is a promising therapeutic strategy, but there is no reliable method for synergism estimation. Although the time-kill curve assay is a gold standard, the method is not appropriate for fast and extensive screening of the synergy. The aim is to optimize the checkerboard method to determine phage-chemical agent interactions, to check its applicability by the time-kill curve method, and to examine whether the synergy can be obtained with both simultaneous and successive applications of these agents. In addition, the aim is to determine interactions of the Pseudomonas phage JG024 with ciprofloxacin, gentamicin, or ceftriaxone, as well as the Staphylococcus phage MSA6 and SES43300 with ciprofloxacin, gentamicin, and oxacillin. The results show that the optimized checkerboard method is reliable and that results correspond to those obtained by the time-kill curve. The synergy is detected with the phage JG024 and ciprofloxacin or ceftriaxone against *Pseudomonas aeruginosa*, and the phage SES43300 with ciprofloxacin against MRSA. The synergy was obtained after simultaneous applications, and in the case of *P. aeruginosa,* after application of the second agent with delay of one hour, indicating that simultaneous application is the best mode of synergy exploitation for therapy. The checkerboard method can be used for thorough clinical studies on synergy and in the future for personalized therapy when infections are caused by multiple resistant bacteria.

## 1. Introduction

The emergence of bacterial multiple- and pan-resistance has accelerated research of new antibacterial agents and strategies for bacterial eradication. Among the unconventional agents, bacteriophages are particularly prominent, and when it comes to new strategies, combining antibiotics with other agents is mostly studied in order to achieve synergism. So far, many phages and chemical agents have been combined and the results are promising. These combinations slow the spread of antibiotic-resistant strains and allow the application of antibiotics in sub-inhibitory doses, which are less harmful to animals and humans [1].

Various methods have been used to determine the effect of combining antibiotics as conventional agents and bacteriophages as unconventional.

The first synergy was noticed by Neter and Clark (1944) [2] for the combination of a staphylococcus phage and penicillin against *Staphylococcus aureus*. Almost a century later, the first experiments conducted in order to examine phage and chemical agent interactions were carried out in vivo by Huff et al. (2004) [3], using phage and enrofloxacin against *Escherichia coli* in a chicken model. Several years later, Comeau et al. [4] observed that subinhibitory concentrations of individual antibiotics can affect the production of virulent phages infecting *E**. coli* and the combination was later applied to more efficiently reduce cell densities than either treatment alone. The authors defined the phenomenon of synergy as “phage-antibiotic synergy” or PAS. Among the available methods for synergy determination, a gold standard is the time-kill method, which was applied to confirm phage-antibiotic synergy for the first time by Knezevic et al. (2013) [5].

Phage-antibiotic synergy has been widely examined and confirmed for various phage-antibiotic combinations (reviewed in [1]). However, the methods used for assessment of these interactions are not defined and thus, the methodology used varies significantly. The broth microdilution limited series method, agar dilution method, or disk-diffusion method are used to determine the synergistic effect between bacteriophages and antimicrobial agents [6,7]. Using the disk-diffusion method, the interaction between two agents can be qualitatively checked, most often for antagonism. On the other hand, the broth microdilution checkerboard method is a two-dimensional, two-agent broth microdilution assay for testing the effect of chemical agents against a given microorganism [7,8,9,10,11]. With prior knowledge of the MIC values of the given agents, the checkerboard method is set up in order to check the bacteriostatic and/or bactericidal activity of antimicrobial agents. The detected in vitro interactions are calculated and interpreted as synergistic, additive, indifferent, or antagonistic depending on whether the antibacterial activity of agents in combination is greater than, equivalent to, or less than the activities of agents when applied alone.

Among Gram-negative bacteria, *Pseudomonas aeruginosa* is one of the most frequently drug-resistant bacteria, while among Gram-positive it is *S. aureus* [12]. In addition, phages infecting these two species are well characterized, such as Pseudomonas phage JG024 (species *Pseudomonas virus JG024*; genus *Pbunavirus*; class *Caudoviricetes*) and Staphylococcus phage MSA6 (species *Staphylococcus virus MSA6*; genus *Kayvirus*; family *Herelleviridae*; order *Thumleimavirales*; class *Caudoviricetes*). Accordingly, these bacteria-phage systems are excellent models for synergy examination.

Since there is a necessity for phage-chemical agent synergy detection, and there are no standardized, reliable, as well as fast and easy to perform methods, the aim of the study was to adjust a checkerboard method for this purpose and check the results using a gold standard time-kill curve method in parallel. In addition, we examined effect of combination of selected *P. aeruginosa* and *S. aureus* phages and antibiotics in simultaneous and successive treatment, to additionally elucidate the phenomenon.

## 2. Materials and Methods

### 2.1. Bacterial Strains

*P. aeruginosa* reference strain UCBPP-PA14 (PA14) and methicillin-resistant *S. aureus* ATCC 43300 obtained from American Type Culture Collection were used in this study. Mueller-Hinton (MH) broth (Torlak, Republic of Serbia) was used to store the strains, with 20% glycerol at −80 °C. The strains were propagated in MH broth at 37 °C for 24 h or on Mueller-Hinton (MH) agar, to obtain colonies.

### 2.2. Bacteriophages

Pseudomonas phage JG024 was obtained from The German Collection of Microorganisms and Cell Cultures (Deutsche Sammelung von Mikroorganismen und Zellkulturen, Leibniz Institute, Berlin, Germany). A staphylococcus phage designated as SES43300 was isolated from a commercial SES preparation (Eliava Institute, Tbilisi, Georgia) and Staphylococcus phage MSA6 was obtained from Dr. Lidia Mizak (MIHE—Military Institute of Hygiene and Epidemiology, Warsaw, Poland). The phages were propagated using double layer method, precipitated with PEG6000, purified by CsCl equilibrium ultracentrifugation and dialyzed, as described previously [13]. The obtained phage titer was determined by SPOT method and double layer method. The prepared phage stocks were stored at 4 °C or at −80 °C with 10% glycerol.

### 2.3. Antimicrobial Agents

Ciprofloxacin (CIP), gentamicin (GEN), oxacillin (OXC) and ceftriaxone (CRO) (Sigma, Aldrich, St. Louis, MO, USA) were used as antibiotic solutions.

### 2.4. Minimal Inhibitory Concentrations (MICs)

Minimal inhibitory concentrations (MICs) for antibiotics were determined for *S. aureus* and *P. aeruginosa* strains using broth microdilution method in 96-well plates. Prepared bacterial suspensions of 0.5 McFarland density (~10^8^ CFU mL^−1^) of overnight cultures were diluted in MH broth (1:100 *v*/*v*). Then, 100 μL of an inoculated medium was added to the wells of a microtiter plate. The same volume of antibiotics was added to the wells, with the final concentration of each antibiotic in the microtiter plates ranging from 0.0625 to 128 μg mL^−1.^ The following antibiotics were used: CIP, GEN and CRO, and for *S. aureus* OXC instead of CRO. Total bacterial counts in each well of the microtiter plate were ~ 1 × 10^6^ CFU mL^−1^. The microtiter plates were incubated for 18–24 h at 37 °C, after which 20 μL of 0.1% solution of 2,3,5-triphenyltetrazolium chloride (TTC) was added to each well. The microtiter plates were further incubated for 2–3 h at 37 °C to allow bacteria to convert TTC to red formazan by the dehydrogenases of viable bacterial cells, after which the MIC value for each antibiotic was determined. The lowest concentration of antibiotics needed to prevent formazan production was considered as an MIC. The reference strain *S. aureus* ATCC 25923 was used as a control. The experiment was performed in triplicate and in at least three independent experiments. The obtained results are presented as geometric mean.

### 2.5. Phage Lytic Efficacy

The phage lytic efficacy of selected bacteriophages was determined by the method described by Knezevic and Petrovic (2008) [14] with certain modifications. Ten-fold dilutions of phages were prepared in SM buffer from phage stocks. Initial phage number was 10^9^ PFU mL^−1^. Overnight bacterial cultures were used to prepare suspensions of 0.5 McFarland in PBS, approximately 2 × 10^8^ CFU mL^−1^. The same volume of double concentrated MHB and phage suspension (1:1) was added to microtiter plate wells. Finally, an inoculated medium (1:100) was added to the wells and the final number of bacteria was 1 × 10^6^ CFU mL^−1^. The total volume in wells was 200 µL and the plates were incubated at 37 °C for 24 h. After the incubation, the multiplicity of infection (MOI) that prevented the bacterial growth was determined and denoted as “minimal inhibitory MOI” (MIM). The MIM was further confirmed by addition of 20 µL of TTC and incubation at 37 °C for 2–3 h, until formation of red formazan. The formazan formation was detected by both visual inspections and by reading the absorbance at 540 nm using a microtiter plate reader (Multiscan GO, Thermo Fisher Scientific, Vantaa, Finland). The MIMs were determined in two replicates and two independent experiments. After this experiment, lytic efficacy was performed with two-fold dilutions from the first higher ten-fold dilution that showed a MIM value. The principle of the experiment is the same as with ten-fold dilutions.

### 2.6. Checkerboard Assay

The bacteriophage–antibiotic interactions were examined using the checkerboard method in the 96-well plate, modified to this specific combination. Each plate was prepared in such a way to vary antibiotic two-fold concentrations in columns (from 1 to 10) and phage MOI in the rows (from A to G). Column 11 contained only antibiotic (Ma), while row H contain only phage (Mf). Column 12 contained controls: bacterial growth (C1), phage suspension sterility (C2), antibiotic stock sterility (C3), and medium sterility (C4) (Figure 1).

The microtiter plates were incubated for 24 h at 37 °C, after which 20 μL of 0.1% solution of TTC was added to each well, which was reduced to red formazan by the dehydrogenases of viable bacterial cells. The experiment was done in triplicate and results were averaged. All combinations of different concentrations of antibiotics and bacteriophages needed to prevent red formazan formation were used to determine Fractional Inhibitory Concentration Index (FICI). FICI is determined using the following formula:FICI = (MICcombi/MICalone) + (MIMcombi/MICMalone),(1)

A FICI of ≤0.50 was defined as synergy, a FICI between 0.50 and 1.00 represents an additive effect, a FICI from 1.00 to 2.00 was defined as indifferent. An FICI above 2.00 represents antagonism [15]. The isobologram with the obtained values was constructed in Origin 2021.

Checkerboard assay was performed in several different ways: simultaneously adding both phage and antibiotic and successively adding one agent and after 1 h or 6 h another agent and vice versa.

#### 2.6.1. Simultaneous Treatment

Two-fold bacteriophage dilutions were prepared in rows of microtiter plates, starting from A1 to A10 and so on to the row H. A bacteriophage suspension corresponding to 10 MOI was added to the first wells. The final volume in each well was 50 µL. H1–H10 was used to determine the MIM on a given lytic phage. Two-fold dilutions of the antibiotic were prepared, and the lowest concentration of antibiotics was added to the G row and in each subsequent row starting from the G to A, the antibiotic concentration was doubled. The final volume in each well was 100 µL. Column A11–H11 was used to determine the MIC on a given antibiotic. The last column A12 to H12 was used to check the sterility of the medium, the suspension of antibiotics and bacteriophages, and to check the growth of bacteria. Two wells were used for each control. Overnight bacterial culture was centrifugated at 6000 *g*, for 2 min. The supernatant was removed and the pellet was homogenized in PBS buffer. This step was repeated once and the bacteria were used to make 0.5 McFarland suspension density (~10^8^ CFU mL^−1^). Double concentrated MH broth was inoculated with the prepared bacterial suspension (~10^6^ CFU mL^−1^). To each well, 100 µL of inoculated broth was added. The final volume in each well was 200 µL.

#### 2.6.2. Successive Treatment

In successive treatment, one antimicrobial agent is added before inoculation of the bacterium and the other 1 or 6 h later. Both bacteriophages and antibiotics were added to the microtiter plate according to the same template as for simultaneous treatment. Solutions of antimicrobial agent at a 10 times higher concentration or MOI than those used for simultaneous treatment were prepared for successive treatment. After a given incubation period (1 or 6 h), 20 µL of well content was removed (from A1 to H10) and a second antimicrobial agent was added in the same volume to each well. The final volume in each well was 200 µL. Successive treatments were marked as phage + antibiotic or vice versa depending on which of two agents was added first and which one was added 1 or 6 h later.

### 2.7. Time-Kill Kinetics Assay

The time-kill kinetics assay was used to check the accuracy of the new optimized checkerboard method. The bacterial suspension was prepared in the same way as for the checkerboard method and used for inoculation of double concentrated MH broth (~10^6^ CFU mL^−1^). The bacterium was treated simultaneously and successively after 1 and 6 h with 1/4 MIC for the given antibiotic and 1/4 MIM for bacteriophage. The bacterium was also treated individually with 1/4 MIC of the antibiotic and with 1/4 MIM of the bacteriophage. Negative control was the uninoculated medium and the final volume in all tubes was 10 mL. In order to monitor the growth of bacteria in simultaneous and successful treatments, the total number of bacteria in each tube was determined at the beginning of treatment and every 3 h using the spread plate method. After 24 h of incubation on 37 °C, agar plates were observed and colonies counted in order to determine total bacterial number, used to construct time-kill curves. The experiment was done in triplicate and the results were averaged. Phage antibiotic synergism obtained through time-kill is implied by reduction in bacterial number when treated with combination of subinhibitory concentrations for two or more log, comparing to the number of bacteria treated with only one, more effective agent. Likewise, synergism implies the reduction in bacterial number regarding the initial bacterial number.

## 3. Results

The obtained MIC and MIM values, used as a basis for the synergy experiments, are shown in Appendix A.

### 3.1. Simultaneous Treatment

#### 3.1.1. *Pseudomonas aeruginosa*

The combinations of CIP, GEN, or CRO with JG024 phage were investigated during simultaneous treatment (Figure 2). A synergistic effect was found between CIP and JG024 (FICI = 0.29–0.50) as well as CRO with JG024 (FICI = 0.18–0.50). On the other hand, simultaneous treatment of GEN and JG024 on PA14 strain was not effective (FICI = 1.00–1.83).

The effect of simultaneous antibiotic-phage treatment was also tested by the time-kill method. CIP in combination with JG024 showed a synergistic effect on the growth of PA14 strain (Figure 2). After 24 h of incubation, the decrease in the bacterial number compared to the initial one was 2.01 log. Simultaneous treatment with a combination of CRO and JG024 phage was also effective. This combination caused a decrease in the bacterial number of 2.65 log after 24 h of incubation in comparison to the bacteria treated with only one of these agents. Finally, GEN, as in the case of the checkerboard method, was not effective in combination with the JG024 phage and the bacterial count barely differed from the number of bacteria treated with only GEN or JG024, as the combination only reduced bacterial count for 0.05 log comparing to the single agent treatments. The results of the time-kill method match the results of the checkerboard method.

#### 3.1.2. *S. aureus*

During simultaneous treatment, different combinations of CIP, GEN, and OXC were used with MSA6 or SES43300 phages against *S. aureus*. A synergistic effect was only detected between CIP and SES43300 (FICI = 0.42) and confirmed by the time-kill method (Figure 3). CIP with MSA6 phage showed FICI = 0.85–1.33 (the results are not shown). OXC with phage SES43300 showed an additive effect (FICI = 0.56–1.2), while GEN showed similar (FICI = 1.03–1.75) (Figure 3). MSA6 showed an indifferent effect with OXA (1.03–2.00) and additive or indifferent effect with GEN (FICI = 0.88–1.25) (the results are not shown).

### 3.2. Successive Treatment

#### *P. aeruginosa* 

Successive treatment with CIP and JG024 also showed a synergistic effect (Figure 4). When the JG024 phage is added to the CIP treatment with a delay of 1 h, the FICI was 0.21–0.50, while after adding CIP to the phage treatment 1 h later, the FICI was 0.26–0.50.

The time-kill method confirmed the synergistic effect of successive treatment in both cases after 1 h (Figure 4). After adding JG024 1 h later, the total number of bacteria at the end of incubation was 4.28 ± 0.82 log, which is 2.08 log less than the initial number. A similar effect was achieved after the addition of CIP 1 h later and the decrease in the combined treatment compared to the treatments with only one agent was 2.09 log. The results of the time-kill method match the results of the checkerboard method.

However, successive treatment when the second antimicrobial agent was added 6 h after the addition of the first one was not effective (Figure 5). Neither the addition of the phage as a second agent nor CIP after 6 h inhibit bacterial growth, with FICI 1.00–2.00. The time-kill curves obtained with delayed treatment after 6 h confirmed the lack of synergy (Figure 5).

Unlike successive treatment after 1 h, treatment after 6 h in the case of CIP and JG024 did not significantly reduce the number of bacteria at the end of incubation (Figure 5). After 24 h, the total number of bacteria in the combination CIP + JG024 is 6.62 ± 1.11 log, and in the case of JG024 + CIP 7.48 ± 0.24. The results of the time-kill method match the results of the checkerboard method.

JG024 + GEN and GEN + JG024 after 1 h of addition did not have a synergistic effect. Successive treatment of GEN with JG024 in combination when one antimicrobial agent is added after 1 h of incubation was not effective in lowering the total bacterial number (Figure 6). In the combination GEN + JG024, the total number of bacteria was 7.28 ± 1.46, and in JG024 + GEN 7.29 ± 0.5. The synergistic effect between GEN + JG024 was not determined by the checkerboard method, which coincides with the time-kill method. In the case of JG024 + GEN, the lack of synergy was confirmed by both checkerboard and time-kill curve method.

An indifferent effect of GEN and JG024 was determined by the checkerboard method in successive treatment where the other agent was added 6 h later. In both cases of successive treatment with GEN and JG024 phage after 1 h and 6 h, an indifferent effect was found. FICI for GEN + JG024 was 0.75–2.00, and for JG024 + GEN 1.00–2.00. The time-kill method confirmed that the addition of one of these two agents after 6 h of incubation did not significantly reduce the growth of the PA14 strain (data not shown). In the combination GEN + JG024 after 24 h of incubation, the total number was 8.15 ± 0.29 log, and for JG024 + GEN 7.05 ± 1.01 log.

The combination where JG024 was added 1 h after incubation of the PA14 strain with CRO was effective against bacterial growth (Figure 7). The synergistic effect was determined and the FICI was 0.08–0.50. However, the addition of CRO antibiotics after 1 h did not have the same effect and the FICI was 1.00–2.00. After 24 h of incubation, the total number of bacteria was 4.23 ± 0.16 log, which is 2.08 log less than the treatments with single agents. In the reverse combination when CRO was added 1 h later, no significant difference was achieved in the total number of bacteria compared to the initial one. The results of the time-kill method match the results of the checkerboard method.

As in the case of ciprofloxacin and gentamicin with JG024 in successive treatment where one antimicrobial agent is added 6 h after incubation, the combination of ceftriaxone with this phage was also unsuccessful. The FICI for the combination CRO + JG024 was 1.00–2.00, and for the combination JG024 + CRO, the FICI was the same, 1.00–2.00 (data not shown).

Successive treatment after 6 h in combination with CRO with JG024 did not make a difference of 2 log compared to the initial bacterial number. The same case was in combinations of JG024 phage with CIP and GEN. The combination of CRO + JG024 after 6 h had a total number at the end of incubation of 5.31 ± 1.55, and in the case of JG024 + CRO 5.60 ± 1.73. The results of the time-kill method match the results of the checkerboard method (the results are not shown).

In successive treatment of MRSA strain ATCC43300, regardless to the combined phages (MSA6 or SES43300) and antibiotics (CIP, GEN or OXC) after 1 or 6 h, a synergy was not obtained (the results are not shown).

## 4. Discussion

Phage-antibiotic synergy offers a promising therapeutic solution for eradication of multiple- and pan-resistant bacteria. The previous results showed that the phenomenon is dependent on antibiotic, phage, and bacterial strain [1], so PAS should be determined for each specific combination. However, methods for PAS significantly vary and it is impossible to compare results obtained by different authors. Moreover, the PAS is rarely confirmed by the time-kill curve method, which is a gold standard in synergy examination. The reason why time-kill curve is rarely used is that the experimental procedure is robust and complex, requires work with larger volumes, and constant determination of the bacterial number in short time-intervals. When interactions of chemical agents are tested, for a preliminary estimation of interactions, a checkerboard method is used, so here we adjusted the method for phage-antibiotic combinations and examined its applicability for PAS testing.

The modified method comprises phage application as various MOIs, and antibiotic as various concentrations. Such approach allows a FICI calculation without a unit, i.e., an index is obtained. Calculation of FICI is in accordance with standard methods for determining synergism between agents, which is an advantage compared to the synogram proposed by Liu et al. (2020) [16], based on the calculation of growth reduction and the result presentation in the form of a matrix-like heat map. In the experiments, we used two-fold dilutions of both antibiotic concentration and phage MOI to enhance precision, although satisfactory results were obtained with phage ten-fold dilutions (results are not shown). The choice of dilution depends on the lytic efficiency of the phage, so knowing the Virulence Index [17] and PhageScore [16,18] can be very helpful. Notably, the PhageScore method itself has the potential to be developed as an additional method for synergy estimation. The microtiter plate is organized in such a way to have additional control of phage MIM (row H) and antibiotic MIC (column 11), so the interpretation of the results is more reliable and accurate, as it eliminates possible variation of MIC or MIM among experiments. The plate also comprises controls of bacterial growth, as well as antibiotic, phage suspension, and medium sterility, which eliminates possible errors due to contamination.

The results obtained by the checkerboard method were confirmed by a time-kill curve for both *P. aeruginosa* and *S. aureus*. In all cases, regardless to a simultaneous or successive application of phages and antibiotics, when synergy was detected by the checkerboard method, the bacterial number reduction with combination in time-kill test was ≥2 logs in comparison to a more active single agent. This implies that the checkerboard method can not only be used for preliminary estimation of phage–antibiotic interactions against particular bacterial strain, but can also be a very reliable method that correlates with the time-kill curve assay. The results can be determined without the addition of TTC, but formazan formation facilitates obtaining results.

The results show that both phages can be used in combination with antibiotics. Among the examined chemical antimicrobials, CIP seems to be the most promising for PAS. For *S. aureus,* it is of particular interest, as it was confirmed that MRSA shows a high degree of resistance to fluoroquinolones, especially CIP [19]. The obtained results are in accordance with previous findings that a phage–CIP combination can reduce *S. aureus* resistance to the fluoroquinolones [20] and enhance phage production [21]. Previous research showed synergistic effect of staphylococcal phages with GEN [22,23] and OXC, but predominantly against methicillin-sensitive *S. aureus* (MSSA) [24], which was not confirmed in our study. To improve the weak effect of these combinations, a possible solution could be to combine two or more phages with one antibiotic or vice versa [16,17].

The combination of JG024 phage with CIP in simultaneous or successive treatments was effective against the growth of *P. aeruginosa*. These results are in accordance with previously obtained results for Pseudomonas phage GUMS6 [25] and Pseudomonas phage ϕPA01 [26], belonging to the same genus as JG024 phage. As JG024 showed synergism at 0.25 MOI in combination with CIP with eight times lower concentration than MIC (0.03125 μg mL^−1^), this combination seems particularly promising, suppressing bacterial growth for at least 24 h.

Similar to CIP, the combination of CRO with JG024 showed a synergistic effect but only in the case of simultaneous treatment and phage addition after 1 h of incubation against *P. aeruginosa*. Similarly, KPP22 phage, from the same genus, has also been previously tested in combination with antibiotics that inhibit cell wall synthesis using the antibiotic-disk-embedded double-layered agar method synergism was obtained [27]. Both CIP and CRO cause cell filamentation and this can be one of the dominant mechanisms for synergy, as proposed previously [4,28]. Also, other mechanisms should not be neglected, e.g., a decrease in the number of resistant bacterial mutants, as phage lyse cells resistant to antibiotics, and antibiotics kill phage-resistant cells, which causes an effective reduction in bacterial growth [1].

It is interesting to note that the most promising are simultaneous combinations of phages and antibiotics, or successive application of agents, but only after a short delay. If the second agent is applied after 6 h, the synergy was not obtained. The reason is probably that bacteria are multiplied in the presence of subinhibitory MIC/MIM and reach such a number that they cannot be reduced after 6 h, when the second agent is applied. The previous studies showed that the beginning of the log phase and time when bacterial number starts decreasing could affect a lytic potential of the phages, i.e., phages could cause the major bacterial growth inhibition when added at that point of the log phase [29]. This implies that the period when phages should be added depends on the bacterial and phage species that are used in treatment. Liu et al. (2020) [16] point out that the following aspects should also be considered when testing the effects of phages and antibiotics: antibiotic class, bacterial resistance, and various effects of similar phages against the same bacteria.

This study provides a reliable method for the determination of the synergistic effect of phage and chemical agent combinations, which will significantly facilitate future research. Currently, there are some reports on phage–antibiotic interactions in clinical practice (eg., [30]), but a thorough clinical study focused exclusively on this phenomenon has not yet been conducted. Furthermore, the method will contribute to personalized phage therapy with PAS exploitation because combinations of phages and antibiotics against certain clinical strains can be tested relatively quickly and easily.

## Figures and Tables

**Figure 1 viruses-14-01542-f001:**
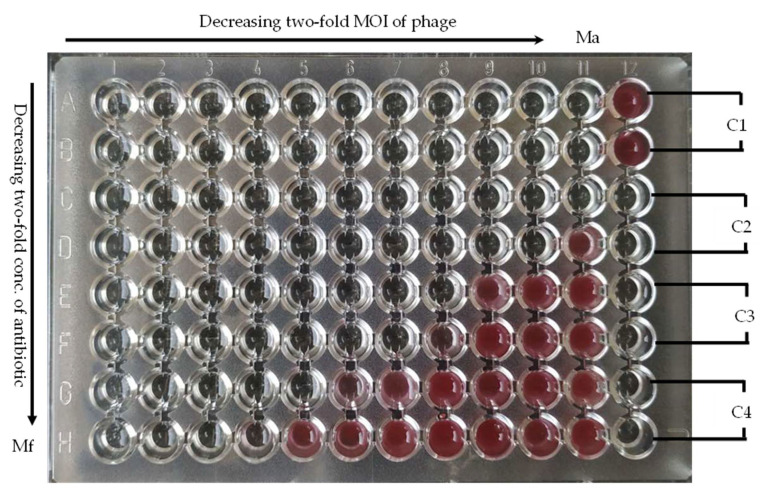
A microtiter plate with checkerboard to estimate interaction between antibiotic and phage (the plate represents combination of CIP and phage JG024 against *P. aeruginosa*). The explanation is in the text.

**Figure 2 viruses-14-01542-f002:**
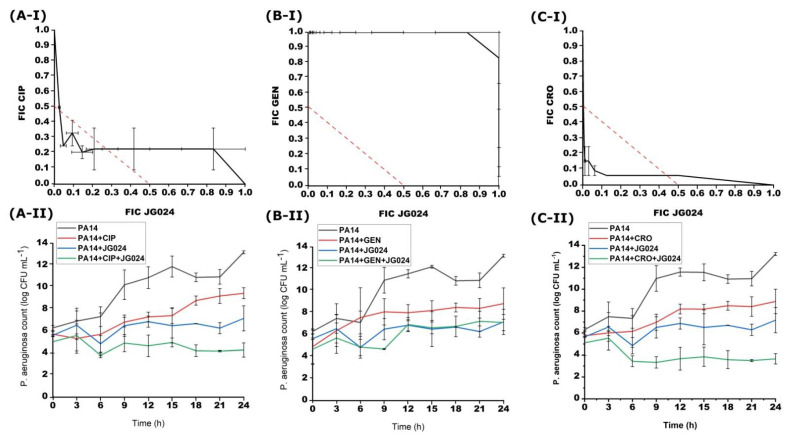
Effects of different simultaneous combinations of JG024 and CIP (**A**), GEN (**B**) or CRO (**C**) on the growth of PA14 strain obtained with checkerboard (**I**) and time-kill method (**II**).

**Figure 3 viruses-14-01542-f003:**
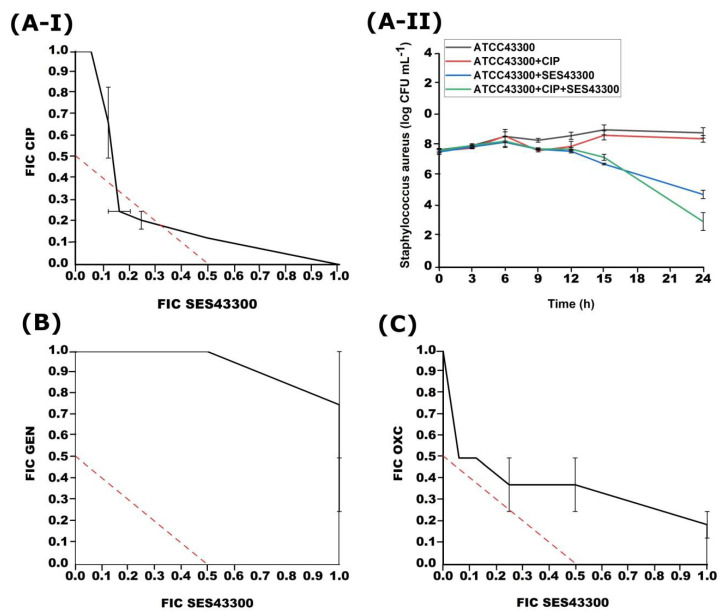
Effects of different simultaneous combinations of SES43300 and CIP (**A-I**), GEN (**B**) or OXC (**C**) on the growth of *S. aureus* ATCC43300 strain obtained with checkerboard and time-kill method for SES 43300 and CIP (**A-II**).

**Figure 4 viruses-14-01542-f004:**
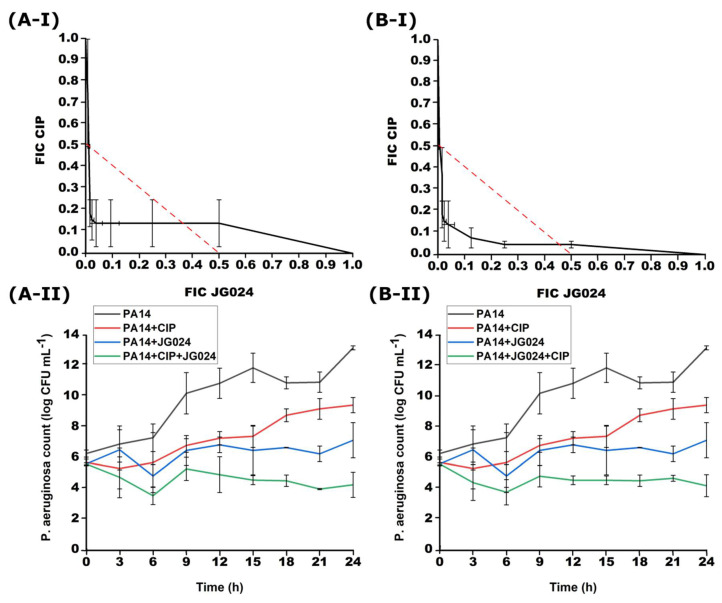
Effect of successive treatment with combination of CIP and JG024 with 1 h delay for each agent obtained with checkerboard (**I**) and time-kill (**II**); (**A**) addition of JG024 1 h later; (**B**) addition of CIP 1 h later.

**Figure 5 viruses-14-01542-f005:**
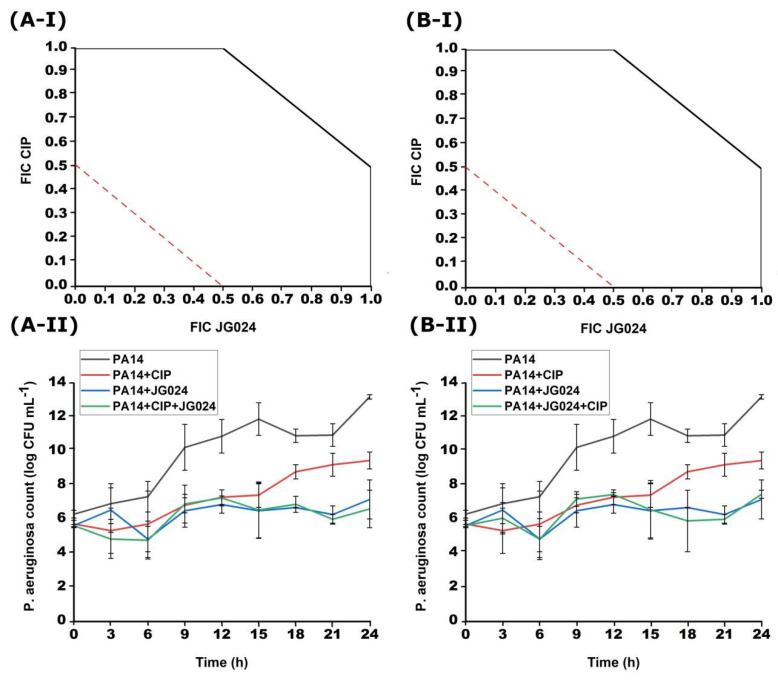
Effect of successive treatment with combination of CIP and JG024 with 6 h delay for each agent obtained with checkerboard (**I**) and time-kill (**II**). (**A**) addition of JG024 6 h later; (**B**) addition of CIP 6 h later.

**Figure 6 viruses-14-01542-f006:**
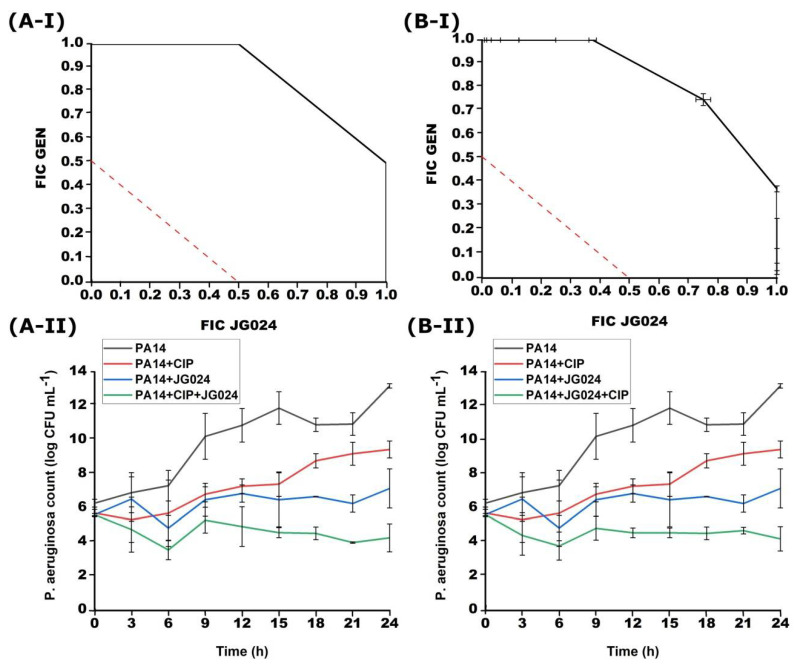
Effect of successive treatment with combination of GEN and JG024 with 1 h delay for each agent obtained with checkerboard (**I**) and time-kill (**II**). (**A**) addition of JG024 1 h later; (**B**) addition of GEN 1 h later.

**Figure 7 viruses-14-01542-f007:**
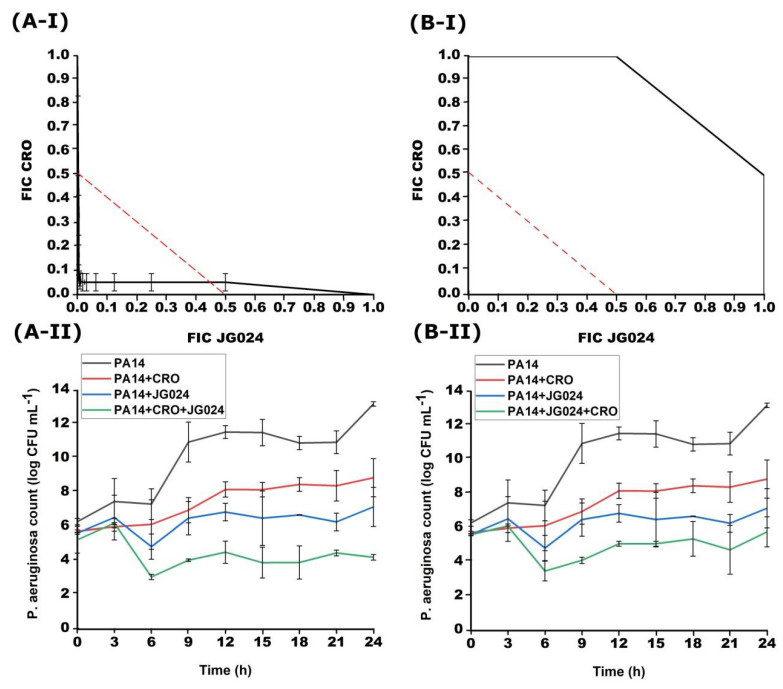
Effect of successive treatment with combination of CRO and JG024 with 1 h delay for each agent obtained with checkerboard (**I**) and time-kill (**II**). (**A**) addition of JG024 1 h later; (**B**) addition of CRO 1 h later.

## Data Availability

Not applicable.

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
