# Peer review of "An Optimized Checkerboard Method for Phage-Antibiotic Synergy Detection"

_viruses, 2022, doi:10.3390/v14071542_

Round 1
Reviewer 1 Report
The manuscript by Nikolic et al. aims at assessing the checker-board method to determine synergistic interactions of a combination of phage and antibiotics against two species of pathogenic bacteria (Pseudomonas aeruginosa and Staphylococcus aureus and phages previously isolated). The investigation is in the line of finding fast and accurate methods to select the best phage-antibiotic pairs for therapies. The experimental design involves exposing bacterial cultures to phages and antibiotics simultaneously and in successive applications and contrasting the check-board results with time-killing assays. Synergy was observed in some combinations of phage-antibiotic when they were applied simultaneously. The author proposed that the checkboard method can rapidly screen synergistic effects of phage-antibiotic combinations for clinical therapies.
Overall, the manuscript presented clear objectives, and the methods are appropriate. The results are of interest to researchers in the field to consider which phage-antibiotic combination is the best for a given infection.
Specific comments.
1. The FICI values reported in the text (l- 204-208) cannot be inferred from figure 2 (and other figures). It seems that the FIC values are normalized from 0 to 1. Please, the authors can clarify these points. Perhaps it is a misunderstanding about how to read the graphs.
2. It is understood that using subinhibitory concentrations of phage makes the synergy pattern emerge in the assays. However, the effect of the combinations seems to be bacteriostatic and not bactericidal. Would you comment on this?
3. The MIC values for antibiotics and MIM for phages should be mentioned in methods or results. They are appropriate references to understand the synergistic effect or lack of synergy, for instance, in the time-killing assays where 1/4 of MIC and MIM were used. If possible, it should be good to comment on the lysis kinetics of the phages at different MOIs.
4. Consider the correction and the suitability of some figures since they are redundant and do not add new information:
Fig. 3 (C). The plot is wrong. It should be FIC SES43300 and FIC OXC in Staphylococcus aureus. Or it corresponds to Fig. 4 C (blank plot).
Fig. 4, Fig. 6, and Fig.10. Consider if these figures are needed. The description of the null effects of the phage MSA6 in the text is enough (Fig. 4). Plots on the six hours successive treatment with phage or antibiotic shown the same effect. Maybe the result can be described in the text.
5. Minor correction: l. 48, Knezevic (2013). Correct.
6. Minor correction: l. 203, l. 223. Cursives.
Author Response
The manuscript by Nikolic et al. aims at assessing the checker-board method to determine synergistic interactions of a combination of phage and antibiotics against two species of pathogenic bacteria (Pseudomonas aeruginosa and Staphylococcus aureus and phages previously isolated). The investigation is in the line of finding fast and accurate methods to select the best phage-antibiotic pairs for therapies. The experimental design involves exposing bacterial cultures to phages and antibiotics simultaneously and in successive applications and contrasting the check-board results with time-killing assays. Synergy was observed in some combinations of phage-antibiotic when they were applied simultaneously. The author proposed that the checkboard method can rapidly screen synergistic effects of phage-antibiotic combinations for clinical therapies.
Overall, the manuscript presented clear objectives, and the methods are appropriate. The results are of interest to researchers in the field to consider which phage-antibiotic combination is the best for a given infection.
Response: Thank you for the comment.
Specific comments.
- The FICI values reported in the text (l- 204-208) cannot be inferred from figure 2 (and other figures). It seems that the FIC values are normalized from 0 to 1. Please, the authors can clarify these points. Perhaps it is a misunderstanding about how to read the graphs.
Response: The FICI values cannot be included in the graph, as each value represent the sum of FIC for an antibiotic (Y-axis), and FIC for a phage (X-axis). The values 1 represent FIC of antibiotic without phage (FICantibiotic=MIC/MIC; X=0, Y=1), and phage without the antibiotic (FIC phage=MIM/MIM; X=1, Y=0). However, we revised the figures and included a dashed line in each isobologram, below which FICI values are >0.5, and represent synergy. The corresponding explanation is included in the text.
- It is understood that using subinhibitory concentrations of phage makes the synergy pattern emerge in the assays. However, the effect of the combinations seems to be bacteriostatic and not bactericidal. Would you comment on this?
Response: Thank you for the comment. It seems that the effect is bacteriostatic in some instances, according to the time-kill curves. Based on a definition, chemical agents have bacteriostatic activity if MIC/MBC>4. (https://pubmed.ncbi.nlm.nih.gov/14999632/). However, each chemical agents, even those that act bactericidal, have bacteriostatic activity in lower concentrations (e.g. ¼ MIC); for example, CIP acts bactericidal, but in the case when applied as 1/4MIC its activity is bacteriostatic. Taking into account phage specific nature, it is impossible to discriminate between bacteriostatic and bactericidal concentrations (as it is impossible to determine MIC and MBC). Thus, even if it can be assumed that synergistic activity is bacteriostatic, we would like to avoid speculation since this cannot be simply applied to phage activity.
- The MIC values for antibiotics and MIM for phages should be mentioned in methods or results. They are appropriate references to understand the synergistic effect or lack of synergy, for instance, in the time-killing assays where 1/4 of MIC and MIM were used. If possible, it should be good to comment on the lysis kinetics of the phages at different MOIs.
Response: The results are included in supplementary Table 1 (MIC) and Table 2 (MIM). The results of MOI effect on lysis were not shown in the manuscript, so it is hard to comment on, but we included consideration of MOI effect on the selection of dilutions (section Discussion).
- Consider the correction and the suitability of some figures since they are redundant and do not add new information:
Fig. 3 (C). The plot is wrong. It should be FIC SES43300 and FIC OXC in Staphylococcus aureus. Or it corresponds to Fig. 4 C (blank plot).
Response: Corrected
Fig. 4, Fig. 6, and Fig.10. Consider if these figures are needed. The description of the null effects of the phage MSA6 in the text is enough (Fig. 4). Plots on the six hours successive treatment with phage or antibiotic shown the same effect. Maybe the result can be described in the text.
Response: We removed Fig. 4 and Fig. 10, but kept Fig. 6 (in Version 2 it is Fig. 5), to have an example of a “negative result”, which also confirms the usability of the checkerboard method.
- Minor correction: l. 48, Knezevic (2013). Correct.
Response: corrected
- Minor correction: l. 203, l. 223. Cursives.
Response: corrected
Reviewer 2 Report
The manuscript of Nikolic et al. describes a checkerboard method to determine phage-antibiotic interactions and compared it to the time-kill curves. The checkerboard and time-kill kinetics assays using reference bacterial strains were performed with Pseudomonas phage JG024 with ciprofloxacin, gentamicin or ceftriaxone, as well as Staphylococcus phage MSA6 and SES43300 with ciprofloxacin, gentamicin and oxacillin, in simultaneous or successive applications strategies. Overall, the results demonstrate that simultaneous application of phage and antibiotic seems to be the best mode to evidence synergy (when existing) between phage and antibiotic. I found the manuscript is well-written and easy to understand. A reasonable amount of work was involved in the study, and as far as I can determine, the results are solid. Seems to me that the method described in this work is reliable and will be a nice resource for determination of the synergistic effect of phage and antibiotic combinations. Thus, in my opinion the manuscript is suitable for publication in Viruses. However, I recommend a few minor corrections in the text:
1. Ln128. Indicate that formation of red formazan is detected by visual inspection.
2. Ln147. Delete the sentence (incomplete duplication).
3. Ln145-150: FICI formula requires a better explanation, to clarify what is MICA and MICB. I suggest reproducing the original FICI formula description presented in reference #15. Perhaps, the authors could define that A= antibiotic and B=Phage or MICB= MIM?
4. Ln203: please use italics
5. Figure 7: upper left panel is missing the label (A-I)
6. Ln286-287: please reword “In both cases of successive treatment after 6 h with GEN and JG024 phage, no difference was found.”
7. Ln360: MSSA?
Author Response
The manuscript of Nikolic et al. describes a checkerboard method to determine phage-antibiotic interactions and compared it to the time-kill curves. The checkerboard and time-kill kinetics assays using reference bacterial strains were performed with Pseudomonas phage JG024 with ciprofloxacin, gentamicin or ceftriaxone, as well as Staphylococcus phage MSA6 and SES43300 with ciprofloxacin, gentamicin and oxacillin, in simultaneous or successive applications strategies. Overall, the results demonstrate that simultaneous application of phage and antibiotic seems to be the best mode to evidence synergy (when existing) between phage and antibiotic. I found the manuscript is well-written and easy to understand. A reasonable amount of work was involved in the study, and as far as I can determine, the results are solid. Seems to me that the method described in this work is reliable and will be a nice resource for determination of the synergistic effect of phage and antibiotic combinations. Thus, in my opinion the manuscript is suitable for publication in Viruses.
Response: Thank you for the comment.
However, I recommend a few minor corrections in the text:
- Indicate that formation of red formazan is detected by visual inspection.
Response: Thank you for noticing this. The formazan formation was detected by both visual inspections and by reading the absorbance at 540 nm using a microtiter plate reader. The manuscript was revised appropriately.
- Delete the sentence (incomplete duplication).
Response: The incomplete sentence was deleted.
- Ln145-150: FICI formula requires a better explanation, to clarify what is MICA and MICB. I suggest reproducing the original FICI formula description presented in reference #15. Perhaps, the authors could define that A= antibiotic and B=Phage or MICB= MIM?
Response: Thank you for noticing this. We included a general FICI equitation in Verison 1, but in Version 2 the equitation is: FICI = (MICcombi/MICalone) + (MIMcombi/MICMalone).
- Ln203: please use italics
Response: Corrected
- Figure 7: upper left panel is missing the label (A-I)
Response: Corrected
- Ln286-287: please reword “In both cases of successive treatment after 6 h with GEN and JG024 phage, no difference was found.”
Response: The sentence was rephrased.
- Ln360: MSSA?
Response: We indicated “methicillin-sensitive S. aureus (MSSA)”.
Reviewer 3 Report
optimization of methods describing phage-antibiotic interaction for planning the therapy poses a hard nut to crack. Properly optimized therapy could have more positive outcomes than not optimized "blind shooting". in my opinion, this work has a large potential and should be published - this is a nice step forward in phage-antibiotic interaction characterization.
for me in the discussion other methods based on calculations should be added - please refer to other methods in Your work. for examples:
[1] Storms ZJ, Teel MR, Mercurio K, Sauvageau D. The Virulence Index: A Metric for Quantitative Analysis of Phage Virulence. Phage 2019;1:17–26. https://doi.org/10.1089/phage.2019.0001.
[2] Liu CG, Green SI, Min L, Clark JR, Salazar KC, Terwilliger AL, et al. Phage-antibiotic synergy is driven by a unique combination of antibacterial mechanism of action and stoichiometry. MBio 2020;11:1–19. https://doi.org/10.1128/mBio.01462-20.
[3] Konopacki M, Grygorcewicz B, Dołęgowska B, Kordas M, Rakoczy R. PhageScore: A simple method for comparative evaluation of bacteriophages lytic activity. Biochem Eng J 2020;161. https://doi.org/10.1016/j.bej.2020.107652.
[4] Grygorcewicz B, Roszak M, Rakoczy R, Augustyniak A, Konopacki M, Jabłońska J, et al. PhageScore-based analysis of Acinetobacter baumannii infecting phages antibiotic interaction in liquid medium. Arch Microbiol 2022;204:421. https://doi.org/10.1007/S00203-022-03020-7.
Author Response
optimization of methods describing phage-antibiotic interaction for planning the therapy poses a hard nut to crack. Properly optimized therapy could have more positive outcomes than not optimized "blind shooting". in my opinion, this work has a large potential and should be published - this is a nice step forward in phage-antibiotic interaction characterization.
Response: Thank you for the comment.
for me in the discussion other methods based on calculations should be added - please refer to other methods in Your work. for examples:
[1] Storms ZJ, Teel MR, Mercurio K, Sauvageau D. The Virulence Index: A Metric for Quantitative Analysis of Phage Virulence. Phage 2019;1:17–26. https://doi.org/10.1089/phage.2019.0001.
[2] Liu CG, Green SI, Min L, Clark JR, Salazar KC, Terwilliger AL, et al. Phage-antibiotic synergy is driven by a unique combination of antibacterial mechanism of action and stoichiometry. MBio 2020;11:1–19. https://doi.org/10.1128/mBio.01462-20.
[3] Konopacki M, Grygorcewicz B, Dołęgowska B, Kordas M, Rakoczy R. PhageScore: A simple method for comparative evaluation of bacteriophages lytic activity. Biochem Eng J 2020;161. https://doi.org/10.1016/j.bej.2020.107652.
[4] Grygorcewicz B, Roszak M, Rakoczy R, Augustyniak A, Konopacki M, Jabłońska J, et al. PhageScore-based analysis of Acinetobacter baumannii infecting phages antibiotic interaction in liquid medium. Arch Microbiol 2022;204:421. https://doi.org/10.1007/S00203-022-03020-7.
Response: Thank you for the valuable suggestion. The references and appropriate considerations were added in the Discussion.